# *In Vitro* and *In Silico* Anti-Picornavirus Triterpene Alkanoic Acid Ester from Saudi Collection of *Rhazya stricta* Decne

**DOI:** 10.3390/metabo13060750

**Published:** 2023-06-13

**Authors:** Maged S. Abdel-Kader, Fahad S. Almutib, Abdullah F. Aldosari, Gamal A. Soliman, Hisham Y. Elzorba, Mohammed H. Alqarni, Reham S. Ibrahim, Hala H. Zaatout

**Affiliations:** 1Department of Pharmacognosy, College of Pharmacy, Prince Sattam Bin Abdulaziz University, Al Kharj 11942, Saudi Arabia; m.alqarni@psau.edu.sa; 2Department of Pharmacognosy, College of Pharmacy, Alexandria University, Alexandria 21521, Egypt; reham.abdelkader@alexu.edu.eg (R.S.I.); hala.zatout@alexu.edu.eg (H.H.Z.); 3College of Pharmacy, Prince Sattam Bin Abdulaziz University, Al Kharj 11942, Saudi Arabia; fahadsmoteb@gmail.com (F.S.A.); ph.abdullah77@gmail.com (A.F.A.); 4Department of Pharmacology, College of Pharmacy, Prince Sattam Bin Abdulaziz University, Al Kharj 11942, Saudi Arabia; g.soliman@psau.edu.sa; 5Department of Pharmacology, College of Veterinary Medicine, Cairo University, Giza 12613, Egypt; elzorba1@gmail.com

**Keywords:** *Rhazya stricta*, picornavirus, foot-and-mouth disease, triterpene alkanoic acid ester, structure elucidation, molecular docking, 3C^Pro^

## Abstract

The total alcohol extract obtained from the aerial parts of *R. stricta* and fractions of the liquid–liquid fractionation process were tested against picornavirus-causing foot-and-mouth disease (FMD) based on the traditional use of the plant in Saudi Arabia. The most active petroleum ether soluble fraction was subjected to chromatographic purification, and nine compounds were isolated, identified using various chemical and spectroscopic methods, and tested for their anti-viral potential. The new ester identified as *α*-Amyrin 3-(3′R-hydroxy)-hexadecanoate (**1**) was the most active compound with 51% inhibition of the viral growth and was given the name Rhazyin A. Compounds with ursane skeleton were more active than those with lupane skeleton except in the case of the acid derivatives where betulenic acid showed 26.1% inhibition against the viral growth, while ursolic acid showed only 16.6% inhibition. Moreover, molecular docking analysis using a glide extra-precision module was utilized for investigating the possible molecular interactions accounting for anti-viral activity against picornavirus of the nine isolated compounds. Molecular docking studies revealed a strong binding of the discovered hits within the active site of FMDV 3C^pro^. Compound **1** showed the lowest docking score within the nine isolated compounds comparable to the two known anti-viral drugs; glycyrrhizic acid and ribavirin. The results of this research will provide lead candidates from natural origin with potential safety and efficacy compared to the synthetic ones with lower production costs for managing FMVD.

## 1. Introduction

*Rhazya stricta* family Apocynacea is a small shrub that grows commonly in the Arabian Peninsula and the Indian subcontinent [1]. *R. stricta* is known in Arabic as “Harmal”. The plant is popular in traditional medicine in many Asian and Middle Eastern countries [2]. In Saudi Arabia, the leaves of *R. stricta* are used as a vermifuge and purgative as well as a treatment for mange [3]. *R. stricta* is used in UAE traditional medicine as an antidiabetic, antihelminthic, anti-inflammatory, skin infections, and stomach disorders [4,5], while in Oman, the leaves of *R. stricta* are used as an antipyretic [6]. In India, the plant is used for the treatment of chronic rheumatism, sore throat, and general debility [7]. Different parts of *R. stricta* are used in Pakistan as a tonic, to cure sore throat, diabetes, constipation, and intestinal and skin diseases [8,9].

The earliest record of probable foot-and-mouth disease (FMD) in cattle was made by Hieronymus Fracastorius in Venice, Italy, in 1514 [10]. Foot-and-mouth disease (FMD) is caused by single-stranded positive-sense RNA picornavirus belonging to the genus Aphthovirus within the family Picornaviridae [11]. FMD is a highly contagious disease that can cause acute and prolonged, asymptomatic, but persistent infection [12]. Susceptible animals are mainly cloven-hoofed animals such as sheep, goats, and cattle [10,11]. FMD virus has seven major serotypes given the symbols: A, O, C, Asia1 and SAT1, SAT2, and SAT3 [13]. The most prominent clinical symptoms of FMD include salivation, loss of body and vesicular lesions on the feet, tongue, and teats, along with fever [14]. The rupture of the vesicles results in marked painful swelling of the coronary band leading to lameness, severe mastitis, and abortions [15]. Due to its serious effects on the livestock industry and international trade in animals and animal products, the Office International des Epizooties (OIE) classified the disease into the A list of infectious diseases of animals [16]. There is no specific treatment for FMD. The conventional method of treating infected animals mainly involves the use of antibiotics, flunixin meglumine, and mild disinfectants [15,17].

In the current study, phytochemical investigation directed by the anti-viral effect against picornavirus was inspired by the common practice in Saudi Arabia among the sheep and cattle herders to use the plant during the outbreak of foot-and-mouth disease. Although we could not find any scientific reference citing this practice, we carried out a detailed study to verify this claim and identify the active components, if any, responsible for this action. The isolated compounds were identified by a combination of spectral and chemical methods. The isolated compounds were further docked against FMDV viral protease, namely 3C^pro^, a necessary target for viral replication. The 3C protease of FMDV (FMDV 3C^pro^) is a chymotrypsin-like cysteine protease, which is one of the most highly conserved proteins among all picornaviruses, including FMDV. This enzyme plays a crucial role in the viral life cycle by cleaving the picornavirus polyprotein into functional mature structural and non-structural proteins [18]. Docking analysis was carried out to investigate the best pose of chosen ligands with the objective protein to gain molecular perception of the inhibitor’s mechanism of binding.

## 2. Materials and Methods

### 2.1. General

Infrared (IR) spectra were recorded on an FT-IR spectrophotometer (Perkin Elmer, Waltham, MA, USA) in the form of KBr pallets. ^1^H, ^13^C-NMR, and 2D-NMR data were collected on a Bruker UltraShield Plus 500 MHz spectrometer (Fällanden, Switzerland) located at the NMR Unite, College of Pharmacy, Prince Sattam Bin Abdulaziz University. The instrument operates at 500 MHz for protons and 125 MHz for carbon atoms, respectively. Chemical shift values were reported in δ (ppm) relative to the residual solvent peaks. Coupling constants (J) were reported in Hertz (Hz). The 2D-NMR experiments (COSY, HSQC, HMBC, H2BC, NOESY, and/or ROESY) were performed utilizing the standard Bruker program. HRMS were determined by direct injection using the Thermo Scientific UPLC RS Ultimate 3000-Q Exactive hybrid quadrupole-Orbitrap mass spectrometer that combines the high-performance quadrupole precursor selection with high resolution and accurate-mass (HR/AM) Orbitrap™ detection (Thermo Fisher Scientific, Waltham, MA, USA). Direct infusion of isocratic elution Acetonitrile/Methanol (70:30) with 0.1% formic acid was used for flushing the samples. Run time was 1 min using nitrogen as auxiliary gas with a flow rate of 5 µL/min. A scan range from 160 to 1500 *m*/*z* was used. Resolving power was adjusted to 70,000 at *m*/*z* 200. Detection was in both positive and negative modes separately. Calibration was done using Thermo Scientific Pierce™ LTQ Velos ESI Positive Ion Calibration Solution including Caffeine, Met-Arg-Phe-Ala (MRFA), Ultramark 1621, n-Butyl-amine components, and Pierce™ LTQ Velos ESI Negative Ion Calibration Solution including sodium dodecyl sulfate (SDS), sodium taurocholate, Ultramark 1621 components. The capillary temperature was set at 320 °C and the capillary voltage at 4.2 Kv. Freeze drying was conducted using a Millroch freeze drier model LD85 (Millroch, Kingston, NY, USA). MPLC was done using a Buchi medium pressure system composed of Buchi pump module C-605 controlled by Buchi control unit C-620 equipped with Buchi fraction collector C-660. The column eluate was detected by Buchi UV photometer C-640. Column 15/460-044032 was used, and the system was operated by Sepacore control chromatography software. Sephadex LH20 (Sigma–Aldrich, Burlington, MA, USA) and silica gel 60/230–400 mesh (EM Science) were used for column chromatography. TLC was done using silica gel 60 F254 (Merck). Centrifugal preparative TLC (CPTLC) using a 4 mm silica gel P254 disc was performed on a Chromatotron (Harrison Research Inc., Union, NJ, USA, model 7924).

### 2.2. Plant Materials

The plants of *Rhazya stricta* Decne, family Apocynacea, were collected in January 2019 from the Al-Kharj region South of Riyadh, 24.20450° N, 47.23455° E, Saudi Arabia. The plant was authenticated by Dr. Mohammad Atiqur Rahman, a taxonomist at MAP-PRC, College of Pharmacy, King Saud University, Riyadh, Saudi Arabia. A voucher specimen (#16723) was kept at the herbarium of this center [19].

### 2.3. Extraction and Fractionation

The air-dried powdered aerial part (1750 g) was extracted with 95% ethanol (4 × 10 L) at room temperature. The extract was evaporated under reduced pressure using a rotary vacuum evaporator to produce 150.92 g of the total extract (RST). The total extract was suspended in ethanol/H_2_O mixture (2:1) and subjected to liquid–liquid fractionation using petroleum ether, CHCl_3,_ and EtOAc to yield 22.10 g of the petroleum ether soluble fraction (RSP), 46.6 g of the CHCl_3_ soluble fraction (RSC), 17.11 g of the EtOAc soluble fraction (RSE), and 62.11 g of the aqueous soluble fraction (RSH).

### 2.4. Chromatographic Purification

Twenty grams of the petroleum ether soluble fraction were subjected to chromatographic purification on a silica gel column (400 g, 5 cm i.d.), eluting with hexane followed by hexane/EtOAc mixtures in a gradient system. Fractions of 150 mL were collected and screened with TLC, and similar fractions were pooled to produce eight fractions A–H.

Fraction B (3.3 g) eluted with 5% EtOAc in hexane was repurified on a silica gel column (150 g, 2.5 cm i.d.) and hexane/EtOAc mixtures in a gradient system. Fractions of 20 mL each were collected and screened by TLC, and similar fractions were collected. Fractions eluted with 5% EtOAc in hexane (500 mg) were subjected to CPTLC using a 4 mm silica gel P254 disc and hexane/acetone mixture 9:1 as a mobile phase to obtain 40 mg of **1** and 200 mg of **2**. Fractions eluted with 10% EtOAc (435 mg) afforded 120 mg of **3**.

Fraction C (2.3 g) eluted with 10% EtOAc in hexane was purified over RP18 MPLC (45 cm × 1 cm id) eluting with 30% H_2_O in MeOH with increasing MeOH contents in a gradient system till 100% MeOH. Fractions of 15 mL were collected, and similar fractions by TLC were combined. Fractions 46–55 (187 mg) afforded 83 mg of **4**, while fractions 59–67 (110 mg) afforded 43 mg of **5** on crystallization from MeOH.

Fraction D (640 mg) eluted with 15% EtOAc in hexane was crystallized from MeOH to afford 236 mg of **6**.

Fraction D (640 mg) eluted with 15% EtOAc in hexane was further purified over a flash silica gel column (30 g, 1 cm i.d.), eluting with 10% EtOAc in hexane. Fractions of 15 mL were collected, and similar fractions were combined to afford 64 mg of **7**.

Fraction G (2.1 g) eluted with 25% EtOAc in hexane was further purified over a flash silica gel column (100 g, 2.5 cm i.d.) to afford 195 mg of **8** and 58 mg of **9** upon crystallization from MeOH.

Due to the complex ^1^H, ^13^C-NMR spectra of **1** and **2** and the overlapping of signals, it was necessary to carry out some simple chemical reactions followed by spectroscopic measurements to provide undoubtful evidence for structure elucidation.

#### 2.4.1. Acetylation of **1** and **2**

Five mg from **1** and **2** were separately dissolved in 0.5 mL pyridine, 200 μL of acetic anhydride was added, and the mixture was kept in the dark for 24 h. The mixtures were dried under nitrogen to afford chromatographically homogenous products **1a** and **2a**.

#### 2.4.2. Alkaline Hydrolysis of **1** and **2**

Solutions of 10 mg of **1** and 20 mg of **2** in 1 mL of MeOH were stirred with 1 mL of methanolic 0.1 N NaOH for 8 h at room temperature. The mixtures were diluted with 10 mL 0.1 N HCl and extracted with CHCl_3_ (3 × 10 mL). The CHCl_3_ layers were evaporated under reduced pressure and purified over silica gel columns (10 g, 0.5 cm i.d.) eluting with hexane and hexane/EtOAc 95:5. Fractions of 5 mL were collected, and similar fractions were combined to afford 4 mg of **4** and 3 mg **1b** from the hydrolytic products of **1**. The hydrolysis of **2** afforded 11 mg of **5** and 8 mg of **1b**.

#### 2.4.3. Acetylation of **1b**

Two mg of **1b** were dissolved in 100 μL pyridine, and 50 μL of acetic anhydride was added, and the mixture was kept in the dark for 24 h. The reaction mixture was dried under nitrogen to afford chromatographically homogenous products **1bAc**.

### 2.5. Anti-Viral Assay

The anti-viral assay was carried out in the Military Veterinary Hospital, Cairo, Egypt.

#### 2.5.1. Preparations of the Total Extract, Fractions, and Pure Isolates for the In Vitro Assay

The RST, RSP, RSC, RSE, RSH, as well as compounds **1**–**9**, were dissolved in DMSO to obtain 10 mg/mL solutions and filtered using sterile Cobetter syringes with 0.2 µm pore size. The dilution was made using the used culture medium Modified Eagle Medium (MEM).

#### 2.5.2. Determination of Samples Cytotoxicity on BHK (Baby Hamster Kidney) Cells

Micro titer plates 96-wells were seeded with the BHK cells. After the confluent sheet was formed, the growth medium was decanted, and the monolayers of the cells were washed twice with wash media. Different concentrations from the tested samples were prepared, and double-fold dilutions from each sample were made in Modified Eagle Medium (MEM). From each dilution, 100 μL was added to wells in triplicate. Three wells in each plate received a maintenance medium and were kept as a control. The plate was incubated at 37 °C and examined frequently for up to two days. Cells were checked for any physical signs of toxicity, e.g., partial or complete loss of the monolayer, rounding, shrinkage, or cell granulation. A solution of 3-(4,5-dimethylthiazol-2-yl)-2,5-diphenyltetrazolium bromide (MTT) was prepared in 5 mg/mL in Phosphate-buffered saline (PBS) (BIO BASIC CANADA INC). From the MTT solution, 20 μL was added to each well. Plates were placed on a shaking table at 150 rpm for 5 min to thoroughly mix the MTT into the media and incubated at 37 °C under a 5% CO_2_ environment for 1–5 h to allow the MTT to be metabolized. The formed formazan was dissolved in 200 μL DMSO after removal of the growth media and shacked at 150 rpm for 5 min to thoroughly mix the formazan into the solvent. The optical density was measured at 560 nm and subtract background at 620 nm. Optical density should be directly correlated with cell quantity.

#### 2.5.3. MTT Assay Protocol

In 96-well plates, 200 μL media containing 10,000 cells were seeded, leaving three wells empty as blank controls. The plates were incubated at 37 °C, 5% CO_2_ overnight, to allow the cells to attach to the wells. Equal volumes of non-lethal dilution of the tested sample and the picornavirus suspension in sterile double distilled water were incubated for one hour. From this mixture, 100 μL was added to each well, shacked at 150 rpm for 5 min, then the plates were incubated at 37 °C, 5% CO_2_ for one day to allow the virus to take effect. To each well, 20 μL of MTT solution was added, shacked at 150 rpm for 5 min, and incubated at 37 °C under a 5% CO_2_ environment for 1–5 h. The formed formazan was dissolved in 200 μL DMSO after removal of the growth media and shacked at 150 rpm for 5 min to thoroughly mix the formazan into the solvent. The optical density was measured at 560 nm and subtract background at 620 nm.

### 2.6. Molecular Docking Analysis

#### 2.6.1. Preparation of Ligand Structures

The nine isolated compounds were further promoted to molecular docking study to investigate their molecular binding mechanism with the target enzyme. The SDF format of the structure was imported to Schrödinger Maestro 10.2 software package (LLC, New York, NY, USA). To construct the 3D structure and search for alternative conformers, the Lig Prep 2.3 module (Lig Prep, version 2.3, 2015, Schrödinger, Cambridge, MA, USA) was used to perform energy minimization of ligand structures. In order to geometrically optimize each ligand structure and generate tautomers, the OPLS (OPLS 3, Schrödinger, New York, NY, USA) force field was used. Epik was utilized to produce all of the ionization states.

#### 2.6.2. Retrieval and Preparation of Target Protein Structure

Preparation of the target protein structure was carried out with the help of Schrödinger software. To determine protein–ligand binding and interactions, the studied FMDV 3C^pro^ structure was modeled using the published three-dimensional (3D) structure of a wide-type 3C^pro^ [18,20]. The X-ray crystallographic structure of the FMDV 3C protease (PDB: 2WV4) was retrieved from RCSB Protein Data Bank (http://www.rcsb.org/pdb, accessed on 14 April 2023).

The crystal structure of the target protein was downloaded as a PDB file, then prepared and optimized by minimizing the energy utilizing the protein preparation wizard (OPLS 3 force field) module executed in Schrödinger suit. Hydrogen bonds and bond order were assigned after the protein optimization was performed. At pH 7, zero-order bonds to metals and disulfide bonds were also constructed. Additionally, the water molecules were eliminated. For grid box generation, the residues involved in the interactions with the incorporated peptide were utilized to build the grid.

#### 2.6.3. In Silico Molecular Docking

The Glide 10.2 extra-precision module (Glide, version 10.2, 2015, Schrödinger, New York, NY, USA) was used to dock the reduced and refined compounds from the Lig Prep file in extra-precision (XP) mode, with default settings set. The empirical scoring function of the Glide-Dock program was used to create modeling scores. The 2D and 3D ligand-target protein interactions as ion pairs, hydrogen bonds, and hydrophobic interactions have been demonstrated in the Maestro interface to investigate their most preferred binding modes [21,22].

## 3. Results

### 3.1. Compounds Characterization

***α-Amyrin 3-(3′R-hydroxy)-hexadecanoate (Rhazyin A)*** (**1**): White powder; IR (KBr) ν max/cm^−1^ 1729 (C0), 3489 (OH); ^1^H and ^13^C NMR see Table 1 and Table 2; HRESIMS [M+Na]^+^
*m*/*z*: 703.6001 (calcd for C_46_H_80_O_3_+Na, 703.6005).

***3′-Acetyl α-Amyrin 3-(3′R-hydroxy)-hexadecanoate*** (**1a**): ^1^H and ^13^C NMR see Table 1 and Table 2; HRESIMS [M+Na]^+^
*m*/*z*: 745.6100 (calcd for C_48_H_82_O_4_+Na, 745.6111).

***3-Hydroxy-hexadecanoic acid*** (**1b**): ^1^H and ^13^C NMR see Table 1 and Table 2; HRESIMS [M−1]^+^
*m*/*z*: 271.2277 (calcd for C_16_H_32_O_3_-H, 271.2273).

***3-Acetyl-hexadecanoic acid ester*** (**1bAc**): HRESIMS [M−1]^+^ *m*/*z* 313.2388 (calcd for C_18_H_33_O_4_-H, 313.2379).

***lupeol 3-(3′R-hydroxy)-hexadecanoate (Procrim A)***(**2**): White powder; IR (KBr) ν max/cm^−1^ 1735 (C0), 3505 (OH); ^1^H and ^13^C NMR see Table 1 and Table 2; HRESIMS [M+Na]^+^
*m*/*z*: 703.5997 (calcd for C_46_H_80_O_3_+Na, 703.6005).

***3′-Acetyl Procrim A ester*** (**2a**): ^1^H and ^13^C NMR see Table 1 and Table 2; HRESIMS [M+Na]^+^
*m*/*z*: 745.6100 (calcd for C_48_H_82_O_4_+Na, 745.6111).

### 3.2. Anti-viral Assay

#### 3.2.1. Determination of Samples Cytotoxicity on BHK Cells

The maximum non-toxic concentrations [MNTC] were determined for the total extract, fractions, and pure compounds to be used in further biological studies. The results are presented in Table 3.

#### 3.2.2. MTT Assay Protocol

The maximum non-toxic concentrations [MNTC] of the total extract, fractions, and pure compounds were used to determine the anti-viral activity against the picornavirus that causes FMD. The results are presented in Table 4.

### 3.3. Molecular Docking Analysis

Molecular docking analysis of the nine isolated compounds was investigated in an attempt to develop new natural products that can inhibit FMDV replication by targeting a viral protease, namely 3C^pro^. The docking scores and interaction modes were compared with two anti-viral drugs; natural triterpene; glycyrrhizic acid, and a synthetic drug; ribavirin (Table 5).

## 4. Discussion

Many of the currently used drugs were discovered from plants based on their traditional uses. For example, the antimalarial drug artemisinin was discovered from the plant sweet wormwood plant *Artemisia annua* L. used in Traditional Chinese Medicine for the treatment of malaria [23]. It is a common practice in Saudi Arabia among sheep and cattle herders to use *R. stricta* “Harmal” during the outbreak of foot-and-mouth disease to save their animals. Although we could not find any scientific reference citing this practice, we carried out a detailed study to verify this claim and identify the active components, if any, responsible for this action.

The total ethanol extract (RST) and the resulting fractions from liquid–liquid fractionation: petroleum ether (RSP), CHCl_3_ (RSC), EtOAc (RSE), and aqueous fractions (RSH) were all tested for their toxicity on the BHK host cells used to grow the virus. RSE fraction was the most non-toxic fraction with 250 μg/mL MNTC followed by the RSP fraction with 125 μg/mL. The RSC fraction showed the highest toxicity with 7.812 μg/mL MNTC. The experimentally obtained MNTC was used to challenge the picornavirus growth to ensure that the effect is not due to the death of the host cells. The RSP fraction expressed the highest anti-viral activity with 40.1% inhibition of viral growth. Although the RSC and RSH were highly toxic on the BHK host cells, their corresponding MNTC were inactive against the viral growth. Consequently, the active RSP fraction was subjected to comprehensive chromatographic purification to isolate and identify the active secondary metabolites.

Compound **1** showed a complex overlapped ^1^H NMR spectrum (Appendix A, Table 1). Few resolved signals could be assigned, including one broad singlet at δ_H_ 5.16 ppm correlated with the carbon signal at δ_C_ 124.63 ppm (Appendix A, Table 2) assigned for =CH. The pattern of the carbon signals in the ^13^C NMR and DEPT135 experiments indicated a triterpenoidal skeleton. The data of the triterpenoid moiety were closely similar to those reported for α-amyrin [24,25]. However, the H-3 proton appeared as a double doublet with *J* = 4.5 and 11.7 Hz and was downfield shifted to δ_H_ 4.46 ppm compared with the chemical shift values of H-3 in triterpenes (Appendix A). The corresponding C-3 shifted to δ_C_ 80.82 ppm (Appendix A). Both values for H-3 and C-3 were diagnostic for the acylated hydroxyl group at C-3. Due to the overlapping of the proton and carbon spectra in the aliphatic region, it was expected that the aceylating moiety would be a long-chain fatty acid. The multiplet at δ_H_ 3.78 ppm (Appendix A) and the CHOH signal at δ_C_ 70.68 ppm (Appendix A) indicated the presence of a hydroxyl group on the fatty acid part. The presence of a free hydroxyl group in compound **1** was proven by acetylation with acetic anhydride in pyridine to give **1a**. The ^1^H and ^13^C NMR spectra (Appendix A, Table 1 and Table 2) of **1a** pointed out one acetyl group at δ_H_ 1.75, δ_C_ 20.52, and 169.37 ppm. The carbonyl ester was shifted to δ_C_ 169.57 ppm. The CH-O proton signal expressed a downfield shift to δ_H_ 5.46 ppm as a result of the acetylation process. The CH-O proton showed a COSY correlation with the resolved CH_2_ protons at δ_H_ 2.44 and 2.56 ppm (Appendix A). These protons were correlated in the HSQC experiment (Appendix A) to the methylene carbon signal at δ_C_ 39.79 ppm and showed 2-bonds HMBC correlations (Appendix A) with the ester carbonyl at δ_C_ 169.57 and the hydroxyl bearing CH at δ_C_ 70.60 ppm giving the hydroxyl group position 3′ on the fatty acid skeleton and the CH_2_ position 2′. This assignment was further supported by the two-way H2BC correlations (Appendix A) between the C-2′ CH_2_ and the C-3′ CH-O. To obtain further clear spectroscopic evidence for the structure of compound **1,** it was subjected to alkaline hydrolysis using 0.1 N alcoholic NaOH to hydrolyze the ester group. Hydrolytic products were purified on a silica gel column to obtain the two components of **1**. The triterpene part spectral data were similar to compound **4** and identified as α-amyrin [24,25] by comparison of the obtained spectral data with the literature values. The other hydrolytic product was **1b**, representing the esterifying fatty acid. The spectral data of **1b** were in complete agreement with the 3-hydroxy long-chain fatty acid. Similar correlations were observed in COSY, HMBC, and H2BC between CH_2_ at C-2 and CHOH at C-3 (Appendix A). The length of the fatty acid carbon chain was obtained from the HRESIMS of the fatty acid **1b**, the acetylation product **1bAc**, **1,** and **1a** (Appendix A). HRESIMS of **1b** showed [M−1]^+^ at m/e 271.2277 for the molecular formula C_16_H_32_O_3_-H (Appendix A), while the mono-acetate derivative showed [M−1]^+^ at *m*/*z* 313.2388 for C_18_H_33_O_4_-H (Appendix A). These data enable the identification of the esterifying fatty acid as 3-hydroxyhexadecanoic acid (3-hydroxypalmitic acid). The spectrum of **1** showed [M+Na]^+^ at *m*/*z* 703.5995 for C_46_H_80_O_3_+Na (Appendix A), while that of **1a** showed [M+Na]^+^ at *m*/*z* 745.6100 for C_48_H_82_O_4_+Na (Appendix A) all enable the identification of **1** as the new ester α-amyrin 3-(3′-hydroxy)-hexadecanoate given the trivial name Rhazyin A (Figure 1).

A similar treatment of **2** enables the identification of the acid part as 3-hydroxyhexadecanoic acid (3-hydroxypalmitic acid). The triterpene skeleton was identified as lupeol based on the comparison of the recorded data with the literature values [26,27]. Compound **2** was previously reported from Alecrim–Propolis (Brazilian green propolis) and was given the name procrim A (Figure 1) [28].

The known compounds were identified as lupeol acetate (**3**) [27,29], α-amyrin (**4**) [20,21], lupeol (**5**) [27,29], *β*-sitosterol (**6**) [29], ursaldehyde (**7**) [30], betulenic acid (**8**) [31] and ursolic acid (**9**) [30] based on the comparison with the literature data (Figure 1).

Molecular docking of phytochemicals on appropriate drug target(s) of new diseases provides clues about which plants can be targeted for biological activity screening [32]. The 3C protease of FMDV (FMDV 3C^pro^) is a chymotrypsin-like cysteine protease [33,34], which is one of the most highly conserved proteins among all picornaviruses, including FMDV. This enzyme plays a crucial role in the viral life cycle by cleaving the picornavirus polyprotein into functional mature structural and non-structural proteins [35]. FMDV 3C^pro^ processes 10 of 13 cleavage sites on the polyprotein, making this enzyme an attractive target for anti-viral drugs [18]. Docking analysis was carried out to investigate the best pose of chosen ligands with the objective protein to gain molecular perception of the inhibitor’s mechanism of binding. Molecular docking analysis using a glide extra-precision module was utilized for investigating the possible molecular interactions accounting for anti-viral activity against picornavirus of the nine isolated compounds. To determine protein–ligand binding and interactions, the studied FMDV 3Cpro structure was modeled using the published three-dimensional (3D) structure of a wide-type 3Cpro [20]. The FMDV 3Cpro structure was generated based on the deduced amino acid sequence of FMDV by homology modeling with the PDB ID: 2WV4.pdb using the SWISS-MODEL [36]. The nine isolated compounds and the reference drugs; the natural anti-viral triterpene, glycyrrhizic acid [37], and the anti-viral synthetic drugs, ribavirin [38], were rated based on their extra precision docking scores. As a result of the in silico docking experiments, the screened compounds were arranged based on their docking scores (Table 5). The molecular docking studies revealed a strong binding of the discovered hits within the active site of FMDV 3C^pro^. The highest active compound shown by docking XP g score was Rhazyin A (*α*-Amyrin 3-(3′R-hydroxy)-hexadecanoate).

Docking results revealed that Rhazyin A had the lowest binding energy (−5.048 kcal mol^−1^) and most favorable docking poses, followed by procrim A (−4.76 kcal mol^−1^) and then Lupeol acetate (−3.986 kcal mol^−1^) as shown in (Table 5). The top recognized molecule, Rhazyin A, interactions with the enzyme are depicted in Figure 2. Rhazyin A established three hydrogen bonds with GLY161, SER182, and HIE46 residues, which are important for enzyme-tight binding. Additionally, numerous hydrophobic interactions with VAL28, ALA29, ILE30, VAL140, VAL141, LEU142, MET143, ALA160, ALA163, ALA183, and TYR190 residues and also polar interactions with THR27, HIE46, and SER182 residues in a manner comparable to the binding modalities seen in the previously investigated enzyme substrates ensuring successful docking [18]. Compound 1 (Rhazyin A) showed the lowest docking score within the nine isolated compounds comparable to the two known anti-viral drugs, glycyrrhizic acid and ribavirin. The reference drugs, glycyrrhizic acid and ribavirin interactions with the enzyme, were depicted in S1 and S2, respectively. The molecular docking confirmed the in vitro results, where Rhazyin A showed the most activity with 51% inhibition. To our knowledge, this is the first in silico study on the anti-FMDV-3C^pro^ activity of *R. stricta*.

Different classes of triterpenes such as cycloartane, dammarane, lupane, oleanane, and ursane expressed promising anti-viral activity against dengue virus (DENV), human immunodeficiency viruses (HIV), herpes simplex virus (HSV), hepatitis virus (HV), influenza virus (IV), porcine epidemic diarrhea virus (PEDV), and respiratory syncytial virus (RSV) with selectivity index (SI)/Therapeutic index (TI) > 10 [37].

The MNTC of each of the isolated compounds **1**–**9** were tested for the possible inhibitory effect against picornavirus growth. Compound **1** with ursane skeleton was the most active with about 51% viral inhibitory effect, followed by **2**. Both compounds share the presence of the long-chain alkanoic acid moiety. The acetate ester **3** was as active as **2,** indicating that the esterification of the C-3 OH of these compounds potentiates the anti-viral activity. α-Amyrin was more active than lupeol; however, both compounds expressed weak activity. However, the member with lupane skeleton “betulenic acid” was more active than its analog with ursane skeleton “ursolic acid”. *β*-sitosterol was as active as ursaldehyde with ursane type triterepenoidal skeleton.

## 5. Conclusions

Biologically directed phytochemical study of *Rhazya stricta* against picornavirus causing foot-and-mouth disease revealed that the petroleum ether soluble fraction (RSP) was the most active and the least toxic fraction to the host cells necessary for the viral growth. Extensive chromatographic purification of the RSP fraction resulted in the isolation of nine triterpenoid and steroidal compounds. The structures of the compounds were identified by various spectroscopic and chemical methods. Among the isolates, a new alkanoic acid ester Rhazyin A (α-Amyrin 3-(3′R-hydroxy)-hexadecanoate) was the most active in the anti-viral assay with 51% inhibition of the viral growth using the MNTC. The obtained results indicated that the ester moiety at C-3 is essential for potentiating the anti-viral activity. These findings were supported by molecular docking analysis where Rhazyin A showed the most stable conformation among the nine isolated compounds within the active site of FMDV 3Cpro. Molecular docking enabled us to predict potential interactions between the screened phytochemicals and the disease target. The results of this research suggest a potential role of medicinal plants in the management of FMVD infection and provide lead candidates from natural origin with potential safety and efficacy compared to the synthetic ones with lower production costs for managing FMVD.

## Figures and Tables

**Figure 1 metabolites-13-00750-f001:**
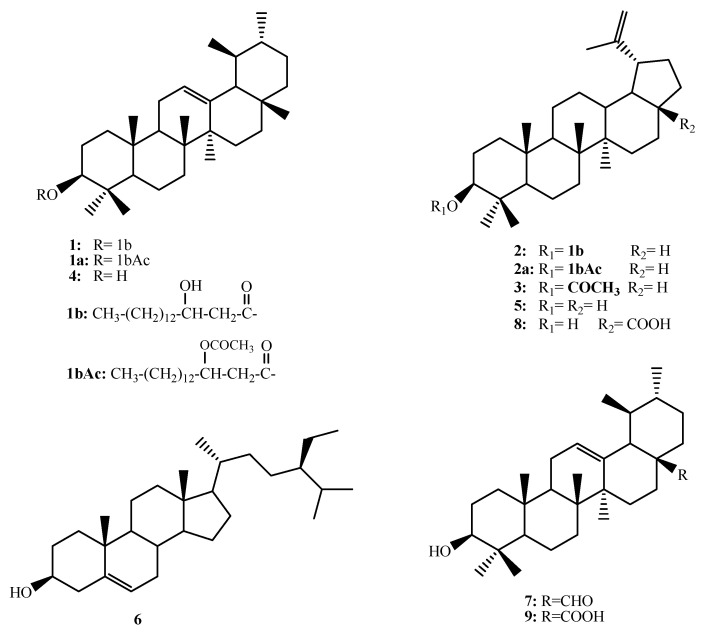
Chemical strucrures of **1**–**9**.

**Figure 2 metabolites-13-00750-f002:**
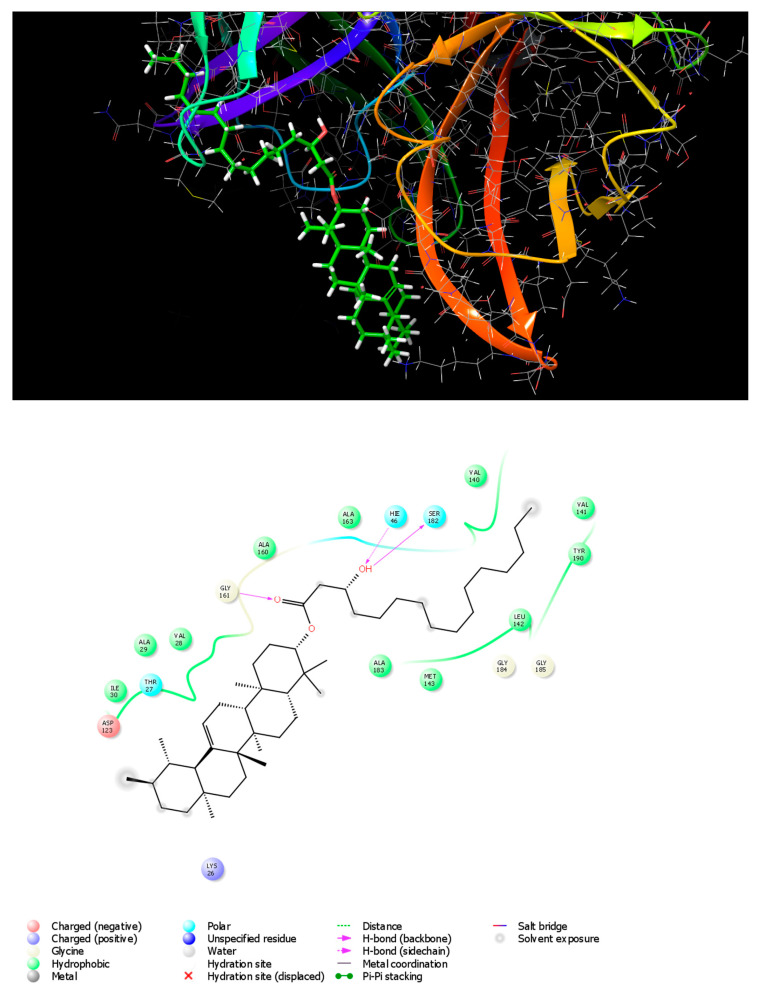
3D and 2D interaction diagrams of FMDV 3Cpro with Rhazyin A.

**Table 1 metabolites-13-00750-t001:** Selected ^1^H-NMR data (δ ppm, *J* in parentheses in Hz) in C_6_D_6_ of compounds **1**, **1a**, **1b**, **2,** and **2a**.

	1	1a	2	2a	1b
**3**	4.46 (dd, 4.5, 11.7)	4.71 (dd, 4.4, 11.8)	4.70 (dd, 4.5, 11.8)	4.72 (dd, 4.5, 11.9)	-
**12**	5.18 bs	5.18 bs	Overlapped	Overlapped	-
**29**	Overlapped	Overlapped	4.73 bs, 4.85 bs	4.74 bs, 4.87 bs	-
**2′**	2.44 (5.1, 15.1) 2.56 (dd, 7.7, 15.1)	2.44 (5.1, 15.1) 2.56 (dd, 7.7, 15.1)	2.33 (d, 6.0)	2.44 (Overlap) 2.56 (dd, 7.7, 15.1)	2.39 (dd, 10.0, 15.9) 2.49 (bd, 15.9)
**3′**	3.78 m	5.46 (p, 6.0)	4.02 m	5.47 (p, 6.8)	3.96 bs
**16′**	Overlapped	Overlapped	Overlapped	Overlapped	0.81 (t, 6.5)
**CH_3_-CO**	-	1.75 s	-	1.71 s	-

**Table 2 metabolites-13-00750-t002:** ^13^C-NMR data (δ ppm) of compounds **1**, **1a**, **1b**, **2** and **2a**.

Pos.	1	1a	2	2a	Pos.	1	1a	2	2a	1b
**1**	38.09	38.02	38.17	38.13	**21**	31.39	31.00	29.50	29.53	
**2**	25.28	27.37	27.52	27.50	**22**	41.67	41.60	40.00	40.00	
**3**	80.82	80.74	80.71	80.71	**23**	27.95	28.38	27.84	27.76	
**4**	36.61	39.94	37.68	38.13	**24**	16.83	16.75	16.00	16.02	
**5**	55.30	55.22	55.27	55.28	**25**	15.61	15.53	16.58	16.58	
**6**	18.26	18.18	18.17	18.16	**26**	16.86	16.78	15.84	15.82	
**7**	32.91	32.83	35.60	35.60	**27**	23.36	23.29	14.50	14.49	
**8**	42.09	42.00	40.75	40.73	**28**	28.83	28.75	17.90	17.91	
**9**	47.61	47.45	50.21	50.17	**29**	17.55	17.47	109.71	109.75	
**10**	37.63	37.25	36.89	36.87	**30**	21.40	21.33	19.20	19.19	
**11**	23.68	23.60	20.80	20.77	**1′**	172.33	169.57	172.33	169.55	177.92
**12**	124.63	124.55	23.76	23.77	**2′**	40.75	39.79	41.89	39.48	41.19
**13**	139.58	139.50	38.03	38.00	**3′**	68.16	70.60	67.98	70.59	68.12
**14**	40.02	42.00	42.76	42.74	**4′**	34.14	34.06	36.81	34.06	36.50
**15**	28.20	28.13	25.19	25.16	**5′**	32.10	32.02	32.00	32.02	31.96
**16**	26.79	26.71	34.21	34.18	**6′–14′**	29.56–29.94	29.49–29.82	29.47–29.85	29.49–29.86	24.70–29.74
**17**	33.81	33.74	42.90	42.91	**15′**	22.88	22.81	22.78	22.81	22.73
**18**	59.16	59.08	48.32	48.30	**16′**	14.15	14.08	14.03	14.08	14.17
**19**	39.57	39.49	48.05	48.08	** C ** **H_3_-CO**		20.52		20.52	-
**20**	39.86	39.63	150.34	150.37	**CH_3_-CO**		169.37		169.34	-

**Table 3 metabolites-13-00750-t003:** Determination of total extract and fractions of *R. stricta* cytotoxicity [MNTC] on BHK cell *.

ID	Dilution 1:2	Viability %	Toxicity %	CC_50_ (ug/mL)	MNTC
**BHK**	ug/mL	100	0	-	
**RST**	1000	14.4	85.6	342.4	62.5
	500	27.1	72.9		
	250	53.8	46.2		
	125	94.1	5.9		
	62.5	100.0	0		
	31.25	100.0	0		
	15.625	100.0	0		
	7.812	100.0	0		
**RSP**	1000	22.6	77.4	653.3	62.5
	500	22.6	77.4		
	250	52.4	47.6		
	125	95.2	4.8		
	62.5	100.0	0		
	31.25	100.0	0		
	15.625	100.0	0		
	7.812	100.0	0		
**RSC**	1000	8.8	91.2	46.1	7.8
	500	9.1	90.9		
	250	17.8	82.2		
	125	26.4	73.6		
	62.5	35.6	64.4		
	31.25	54.3	45.7		
	15.625	90.8	9.2		
	7.812	100	0		
**RSE**	1000	25.3	74.7	699.2	250
	500	60.1	39.9		
	250	99.6	0.4		
	125	100.0	0		
	62.5	98.4	1.6		
	31.25	99.9	0.1		
	15.625	101.0	0		
	7.812	100.0	0		
**RSH**	1000	21.1	78.9	593.0	62.5
	500	46.1	53.9		
	250	77.3	22.7		
	125	91.0	9.0		
	62.5	100.0	0		
	31.25	99.9	0.1		
	15.625	99.5	0.5		
	7.812	100.0	0		

* Raw data are presented in Appendix A.

**Table 4 metabolites-13-00750-t004:** Effect of the total extract, fractions, and pure compounds of *R. stricta* on picornavirus causing FMD *.

Test	Coc. ug/mL	Viability	Toxicity	Viral Activity %	Anti-viral Effect %
**BHK**		100	0		
**FMD**		41.4	58.6	100	0
**RST**	62.5	42.83626	57.16374	97.5	2.5
**RSP**	125	65.0	35.0	59.9	40.1
**RSC**	7.812	38.9	61.1	104.2	0
**RSE**	250	41.1	58.9	100.5	0
**RSH**	62.5	41.2	58.8	100.2	0
** 1 **	500	71.8	28.2	49.0	51.0
** 2 **	125	62.9	37.1	64.5	35.5
**3**	125	62.7	37.3	65	35.0
** 4 **	500	51.9	48.1	83.7	16.3
**5**	250	46.9	53.1	92.4	7.6
** 6 **	250	60.5	39.5	68.7	31.3
**7**	250	60.8	39.2	68.2	31.8
**8**	125	57.5	42.5	74.0	26.0
**9**	125	52	48	83.4	16.6

* Raw data are presented in Appendix A.

**Table 5 metabolites-13-00750-t005:** Free binding energies of the isolated compounds and two reference anti-viral drugs were obtained through molecular docking with FMVD 3C^pro^ crystal structure (2WV4), in addition to types of binding interactions between ligands and critical amino acid residues.

Compound	Docking Score(Kcal mol^−1^)	Interacting Residues	
		Type of Binding Interactions	Amino Acid Residues Involved in Protein–Ligand Interaction
**Glycyrrhizic acid**	−5.894	Hydrogen bonding (backbone)	THR158, GLY161
		Hydrogen bonding (side chain)	HIE46
		Polar interaction	HIE46, THR158, SER182, ASN186
		Hydrophobic interaction	ALA29, ILE30, PRO114, MET143, ALA157, ALA160, TYR162, ALA163, ALA183
		Charged (positive) ionic interaction	ARG159
**Ribavirin**	−5.853	Hydrogen bonding (backbone)	THR158, GLY161
		Hydrogen bonding (side chain)	HIE46, HIE181
		Polar interaction	HIE46, THR158, HIE181, SER182
		Hydrophobic interaction	ALA29, ILE30, CYS31, ALA160, TYR162, ALA163, ALA183
		Charged (positive) ionic interaction	ARG159
**Rhazyin A**	−5.048	Hydrogen bonding (backbone)	GLY161, SER182
		Hydrogen bonding (side chain)	HIE46
		Polar interaction	THR27, HIE46, SER182
		Hydrophobic interaction	VAL28, ALA29, ILE30, VAL140, VAL141, LEU142, MET143, ALA160, ALA163, ALA183, TYR190
		Charged (negative) ionic interaction	ASP123
		Charged (positive) ionic interaction	LYS26
**Procrim A**	−4.76	Hydrogen bonding (backbone)	ILE30, GLY161
		Hydrogen bonding (side chain)	HIE46
		Polar interaction	THR27, HIE46, HIE181, SER182, ASN186
		Hydrophobic interaction	LEU21, VAL28, ALA29, ILE30, CYS31, PRO44, LEU47, MET143, ALA160, TYR162, ALA163, ALA183
		Charged (negative) ionic interaction	GLU50, ASP123
		Charged (positive) ionic interaction	ARG159
**Lupeol acetate**	−3.986	Hydrogen bonding (backbone)	MET143
		Hydrogen bonding (side chain)	
		Polar interaction	HIE46, SER182
		Hydrophobic interaction	VAL28, ALA29, ILE30, LEU142, MET143, ALA160, ALA183, TYR190
		Charged (negative) ionic interaction	GLU50
**Ursaldehyde**	−3.508	Hydrogen bonding (backbone)	ILE30
		Polar interaction	HIE46
		Hydrophobic interaction	LEU21, ALA29, ILE30, CYS31, PRO44, LEU47, MET143, ALA160, ALA163
		Charged (negative) ionic interaction	GLU50, ASP144
***β*-Sitosterol**	−3.503	Hydrogen bonding (backbone)	ILE30
		Polar interaction	HIE46, HIE181, SER182, ASN186
		Hydrophobic interaction	ALA29, ILE30, CYS31, VAL140, VAL141, LEU142, MET143, ALA160, TYR162, ALA163, ALA183, VAL188, TYR190
		Charged (positive) ionic interaction	ARG159
**Betulenic acid**	−3.404	Hydrogen bonding (backbone)	VAL28, GLY184
		Polar interaction	HIE46, SER182
		Hydrophobic interaction	VAL28, ALA29, ILE30, LEU142, MET143, MET148, ALA160, ALA183
		Charged (negative) ionic interaction	GLU50, ASP144, ASP146
		Polar interaction	HIE46, SER182,
		Hydrophobic interaction	LEU21, ALA29, ILE30, CYS31, PRO44, LEU47, ALA60, LEU142, MET143, ALA163, ALA183
		Charged (negative) ionic interaction	GLU50, ASP144
		Hydrogen bonding (side chain)	HIE181
		Polar interaction	HIE46, THR158, HIE181, SER182
		Hydrophobic interaction	VAL28, ALA29, ILE30, MET143, ALA163, ALA160, TYR162, ALA183
		Charged (negative) ionic interaction	GLU50
		Charged (positive) ionic interaction	ARG159
		Glycine interaction	GLY161, GLY184
**Lupeol**	−2.777	Hydrogen bonding (backbone)	VAL28
		Polar interaction	HIE46, SER182
		Hydrophobic interaction	VAL28, ALA29, ILE30, LEU142, MET143, MET148, ALA183, ALA160
		Charged (negative) ionic interaction	GLU50, ASP144, ASP146

## Data Availability

No new data were created or analyzed in this study. Data sharing is not applicable to this article.

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
