# Peer review of "In Vitro and In Silico Anti-Picornavirus Triterpene Alkanoic Acid Ester from Saudi Collection of Rhazya stricta Decne"

_metabolites, 2023, doi:10.3390/metabo13060750_

Round 1

Reviewer 1 Report

The AA report on the effect of a triterpene alkanoic 2 acid ester from Rhazya stricta against Picornavirus causing foot-and-mouth disease. The topic is interested and the research overall well conducted.

However, I have some concerns:

- Can the AA provide some info on the solubility of the extracts? 

- Can the AA provide more details in the Methods section, ie how dilutions/concentrations used for the antiviral activity? How were the extracts dissolved?

- The discussion is weak. The AA mainly report their own results in the discussion section. 

- The Conclusions section could be improved.

Minor points

- please check headings of Table 5

- line 387 (Conclusions section), the word showed is repeated twice.

Some minor editing required.

Author Response

Reviewer 1

We express our thanks to the reviewers for the valuable comments that help improving the quality of the manuscript.

Comments and Suggestions for Authors

The AA report on the effect of a triterpene alkanoic 2 acid ester from Rhazya stricta against Picornavirus causing foot-and-mouth disease. The topic is interested and the research overall well conducted.

However, I have some concerns:

- Can the AA provide some info on the solubility of the extracts? 

- Can the AA provide more details in the Methods section, ie how dilutions/concentrations used for the antiviral activity? How were the extracts dissolved?

Required details were included.

- The discussion is weak. The AA mainly report their own results in the discussion section. 

More discussion was included.

- The Conclusions section could be improved.

Conclusion section modified as recommended.

Minor points

- please check headings of Table 5

Improved as suggested.

- line 387 (Conclusions section), the word showed is repeated twice.

Required changes were performed.

Comments on the Quality of English Language

Some minor editing required.

Linguistic revision was done.

Reviewer 2 Report

Full Title: In vitro and In silico Anti-Picornavirus Triterpene Alkanoic Acid Ester from Saudi Collection of Rhazya stricta Decne.

Manuscript Number: metabolites-2405213

Article Type: Research article

General comments:

In the manuscript “In vitro and In silico Anti-Picornavirus Triterpene Alkanoic Acid Ester from Saudi Collection of Rhazya stricta Decne.”, authors attempted to investigate the Anti-Picornavirus  activity of Rhazya stricta (Apocynaceae) extracts based on the traditional information in Saudi Arabia (KSA). Chemical analysis based chromatography, NMR and in silico molecular docking analysis were used to investigate and validate the traditional uses of Rhazya stricta against Picornavirus causing foot-and-mouth disease.

Overall, the work is original and the authors comprehensively investigated Rhazya stricta. However, the presentation could be improved. The font size and a font theme should be unified and use it then consistently through the whole manuscript. Figures and Tables could be reorganized.

Specific comments:

1.     Line 38–Line 45: Rewrite and rearrange the information in the first paragraph of the introduction part; according to your research work, start with traditional uses in Saudi Arabia, followed by UAE, Oman, India and Pakistan.

2.     Line 51: Start the sentence with the capital letter “Susceptible” instead of “susceptible”.

3.     Line 51–Line 55: Modify the sentence to be more understandable and connected with each other “FMD 52 virus which has 7 major serotypes: A,O,C, Asia1 and SAT 1,2,3 [14] ???”.

4.     Line 78: Write the name of the city and country for the manufacturer company “Infrared (IR) spectra were recorded on FT-IR spectrophotometer (Perkin Elmer) as KBr pallets”

5.     Line 79–Line80: Write the name of the city and the country for the manufacturer company “1H, 13C-NMR and 2D-NMR data were collected on a Bruker UltraShield Plus 500 MHz spectrometer at the NMR Unite, College of Pharmacy, Prince Sattam Bin Abdulaziz”.

8.     Line 79–Line 81: Improve the English language.

9.     Line 90: Split the words “Runtime” to “ run time”.

10.  Line 91: Change the “@ sign” to “at as word” in “Resolving Power was adjusted to 70,000 @ m/z 200. ” to “Resolving Power was adjusted to 70,000 at m/z 200. ”

11.  Line 92–Line 96: Write the name of the city and the country for the manufacturer company “Thermo 92 Scientific Pierce™ LTQ Velos ESI Positive and negative Ion Calibration”

12.  Line 98Line 100: delete the extra space.

13.  Line 97–Line 98: Write the name of the city and the country for the manufacturer company “Freeze drying was conducted using Millroch freeze drier 97 model LD85”

14.  Line 107–Line109: In more details, describe the plant “Rhazya stricta” identification and authentication, without shortening the sentence by refer only to reference No. 20. In addition, it is important to mention the name of the plant family “Apocynaceae” in the text.

15.  Line 118: Change the gram abbreviation “62.11 gm of the aqueous” to “62.11 g of the aqueous

16.  Line 146Line 158: In chromatographic purification, explain why Acetylation and Alkaline hydrolysis methods were specifically used.

17.  Line 165–Line 166: Described in details the antiviral assay method.

18.  Line 180: Space between “150rpm”. Line 304: delete extra space.

19.  Line 193: “for one day to allow the virus to take effect” instead of “for 1- days to allow the virus to take effect”.

20.  Line 209: Move the heading and the first row of Table 1 to the following page.

21.  Line 290–Line294: Improve the text language.

22.  Line 279; Line 325; Line 370: You should choose one format, either without space between the numerical value compounds 1–9 or with spaces between the numerical value “Fractions 46 – 55 (187 mg”), and use it then consistently through the whole manuscript. Also sometimes you have used hyphen (-) as in “Figures S21- S32”; Figures S1-S2 and “1-9”, which it used normally to join the words as in “liquid-liquid fractionation, in Line 115 and Line 286”. However, when showing a range of numerical values, it should be a dash (–).

23.  Line 337: Correct the scientific name Alecrim-Propolisto “Alecrim propolis”, it should be italicized and the genus name start with the small letter, without hyphen between the species and the genus name.

24.  Line 339; Add a space between the reference and the compound number “β-sitosterol (6) [29], ursaldehyde (7) [30], betulenic acid (8) [31] ”.

25.  Line 353: Remove the extra space “rhazyin A (α-Amyrin 3-(3’R-hydroxy)-hexadecanoate)”.

26.  Line 355Line 356: Modify the “kcal mol-1” to “kcalmol−1”.

27.  Line 366: Delete the name of the botanist “Decne”, you have mentioned it already at the beginning at the manuscript.

28.  Line 379Line 390: Adjust the spacing between the lines in conclusion part.

29.  Line 413–Line 480: Unifying italic titles, font size and the use of hyphen and dashes.

Minor editing of English language required

Author Response

Reviewer 2

We express our thanks to the reviewers for the valuable comments that help improving the quality of the manuscript.

General comments:

In the manuscript “In vitro and In silico Anti-Picornavirus Triterpene Alkanoic Acid Ester from Saudi Collection of Rhazya stricta Decne.”, authors attempted to investigate the Anti-Picornavirus activity of Rhazya stricta (Apocynaceae) extracts based on the traditional information in Saudi Arabia (KSA). Chemical analysis based chromatography, NMR and in silico molecular docking analysis were used to investigate and validate the traditional uses of Rhazya stricta against Picornavirus causing foot-and-mouth disease.

Overall, the work is original and the authors comprehensively investigated Rhazya stricta. However, the presentation could be improved. The font size and a font theme should be unified and use it then consistently through the whole manuscript. Figures and Tables could be reorganized.

Specific comments:

  1. Line 38–Line 45: Rewrite and rearrange the information in the first paragraph of the introduction part; according to your research work, start with traditional uses in Saudi Arabia, followed by UAE, Oman, India and Pakistan.

Improved as suggested along with references number necessary changes.

  1. Line 51: Start the sentence with the capital letter “Susceptible” instead of “susceptible”.

Corrected as directed.

  1. Line 51–Line 55: Modify the sentence to be more understandable and connected with each other “FMD 52 virus which has 7 major serotypes: A,O,C, Asia1 and SAT 1,2,3 [14] ???”.

Modified to be more clear.

  1. Line 78: Write the name of the city and country for the manufacturer company “Infrared (IR) spectra were recorded on FT-IR spectrophotometer (Perkin Elmer) as KBr pallets”

Required information were included.

  1. Line 79–Line80: Write the name of the city and the country for the manufacturer company “1H, 13C-NMR and 2D-NMR data were collected on a Bruker UltraShield Plus 500 MHz spectrometer at the NMR Unite, College of Pharmacy, Prince Sattam Bin Abdulaziz”.

Required information were included.

  1. Line 79–Line 81: Improve the English language.

Improved as suggested.

  1. Line 90: Split the words “Runtime” to “ run time”.

Corrected as directed.

  1. Line 91: Change the “@ sign” to “at as word” in “Resolving Power was adjusted to 70,000 @ m/z 200. ” to “Resolving Power was adjusted to 70,000 at m/z 200. ”

Corrected as directed.

  1. Line 92–Line 96: Write the name of the city and the country for the manufacturer company “Thermo 92 Scientific Pierce™ LTQ Velos ESI Positive and negative Ion Calibration”

Required information were included.

  1. Line 98–Line 100: delete the extra space.

Extra spaces were removed.

  1. Line 97–Line 98: Write the name of the city and the country for the manufacturer company “Freeze drying was conducted using Millroch freeze drier 97 model LD85”

Required information were included.

  1. Line 107–Line109: In more details, describe the plant “Rhazya stricta” identification and authentication, without shortening the sentence by refer only to reference No. 20. In addition, it is important to mention the name of the plant family “Apocynaceae” in the text.

More details about the plant collection and identification were included and the family name was mentioned in this section and in the introduction as well.

  1. Line 118: Change the gram abbreviation “62.11 gm of the aqueous” to “62.11 g of the aqueous”

Corrected as directed.

  1. Line 146–Line 158: In chromatographic purification, explain why Acetylation and Alkaline hydrolysis methods were specifically used.

A short paragraph was added to justify the use of chemical reaction beside the spectral data.

  1. Line 165–Line 166: Described in details the antiviral assay method.

Details of the assay are given in 2.5.1.   and 2.5.2.

  1. Line 180: Space between “150rpm”. Line 304: delete extra space.

Corrected as directed.

  1. Line 193: “for one day to allow the virus to take effect” instead of “for 1- days to allow the virus to take effect”.

Corrected as directed.

  1. Line 209: Move the heading and the first row of Table 1 to the following page.

Corrected as directed.

  1. Line 290–Line294: Improve the text language.

Improved as directed.

  1. Line 279; Line 325; Line 370: You should choose one format, either without space between the numerical value “compounds 1–9”or with spaces between the numerical value “Fractions 46 – 55 (187 mg”), and use it then consistently through the whole manuscript. Also sometimes you have used hyphen (-) as in “Figures S21- S32”; Figures S1-S2 and “1-9”, which it used normally to join the words as in “liquid-liquid fractionation, in Line 115 and Line 286”. However, when showing a range of numerical values, it should be a dash (–).

Corrected as directed.

  1. Line 337: Correct the scientific name “Alecrim-Propolis” to “Alecrim propolis”, it should be italicized and the genus name start with the small letter, without hyphen between the species and the genus name.

The name is a type of propolis not organism name, more description was added to clarify.

  1. Line 339; Add a space between the reference and the compound number “β-sitosterol (6) [29], ursaldehyde (7) [30], betulenic acid (8) [31]”.

Corrected as directed.

  1. Line 353: Remove the extra space “rhazyin A (α-Amyrin 3-(3’R-hydroxy)-hexadecanoate)”.

Corrected as directed.

  1. Line 355–Line 356: Modify the“kcal mol-1” to “kcal⋅mol−1”.

Corrected as directed.

  1. Line 366: Delete the name of the botanist “Decne”, you have mentioned it already at the beginning at the manuscript.

Removed as advised.

  1. Line 379–Line 390: Adjust the spacing between the lines in conclusion part.

Corrected as directed.

  1. Line 413–Line 480: Unifying italic titles, font size and the use of hyphen and dashes.

 Corrected as directed.

Comments on the Quality of English Language

Minor editing of English language required

Linguistic revision was done.
